# Infraspecific Variation in *Silene* Seed Tubercles

**DOI:** 10.3390/plants13101416

**Published:** 2024-05-20

**Authors:** José Javier Martín-Gómez, José Luis Rodríguez-Lorenzo, Ana Juan, Ángel Tocino, Emilio Cervantes

**Affiliations:** 1Instituto de Recursos Naturales y Agrobiología, Consejo Superior de Investigaciones Científicas, Cordel de Merinas, 40, 37008 Salamanca, Spain; jjavier.martin@irnasa.csic.es; 2Plant Developmental Genetics, Institute of Biophysics v.v.i, Academy of Sciences of the Czech Republic, Královopolská 135, 61265 Brno, Czech Republic; rodriguez@ibp.cz; 3Departamento de Ciencias Ambientales y Recursos Naturales, University of Alicante, San Vicente del Raspeig, 03690 Alicante, Spain; ana.juan@ua.es; 4Departamento de Matemáticas, Facultad de Ciencias, Universidad de Salamanca, Plaza de la Merced 1-4, 37008 Salamanca, Spain; bacon@usal.es

**Keywords:** curvature, surface geometry, mammillae, seed protuberances, tubercles

## Abstract

The seeds of many species in the order Caryophyllales exhibit surface protuberances called tubercles. While tubercle shape and distribution have often been proposed as taxonomic criteria, paradoxically, their description has primarily relied on adjectives, with quantitative data on tubercle width, height, and other measurements lacking in the literature. Recently, a quantitative analysis of seed surface tubercles based on tubercle width, height, and curvature values (maximum and average curvature, and maximum to average curvature ratio) was proposed and applied to individual populations of a total of 31 species, with 12 belonging to *Silene* subg. *Behenantha* and 19 to *S.* subg. *Silene*. Tubercles were classified into two categories: echinate and rugose. Echinate tubercles exhibited higher values of height and curvature, and lower width, and were more prevalent in species of *S.* subg. *Behenantha*, while the rugose type was more abundant in *S.* subg. *Silene*. This work explored infraspecific differences in tubercle size and shape. For this, measurements of tubercle width, height and curvature were applied to 31 populations of eight species of *Silene*. Significant differences between populations were observed for most of the species examined. A particular tubercle type, previously described as umbonate or mammillate, was identified in *S. nocturna* seeds, characterized by high curvature values.

## 1. Introduction

The Caryophyllaceae Juss. is a family comprising nearly 100 genera and 3000 species, widely distributed in temperate areas of the Northern hemisphere with a centre of diversity in the eastern Mediterranean and Irano–Turanian regions [1,2]. Their seeds develop from anatropous or campylotropous ovules [3], exhibiting a characteristic peripheral position of the embryo [4], and present micromorphological surface protuberances called tubercles. Tubercles with diverse shapes were described in species of many genera of the Caryophyllaceae, including *Arenaria* Ruppius ex L. [5,6,7,8], *Cerastium* Tourn. ex L. [5,9], *Dianthus* [10], *Dichodon* (Bartl. ex Rchb.) Rchb. [6], *Gypsophila* L. [11,12], *Holosteum* Dill. ex L. [6], *Mesostemma* Vved. [6], *Minuartia* Loefl. [5,13], *Moehringia* L. [14], *Paronychia* Hill. [15], *Pseudostellaria* Pax. [6], *Sagina* L. [6,16], *Schizotechium* (Fenzl) Rchb. [6], *Silene* L. [17], *Spergularia* (Pers.) J. Presl & C. Presl [18], *Stellaria* [5,19,20,21,22], and others [6]. Tubercles have also been described in genera of other families of the Caryophyllales Juss., such as the Aizoaceae Martinov [23], Molluginaceae Bartl. [24] and Portulacaceae Juss. [25,26].

The analysis of seed surface and tubercle morphology has traditionally relied on scanning electron microscopy (SEM) because this technique accurately distinguishes between tubercle types and displays their distribution well. However, optical photography provides well-defined images of tubercles, enabling the classification of *Silene* seeds into four groups based on their tubercle type: smooth, rugose, echinate, and papillose [27,28]. Smooth seeds lack tubercles and present the highest values of circularity and solidity in the lateral views. This type predominates in many species of *Silene* sec. *Silene* such as *S. apetala* Willd., *S. colorata* Por., *S. secundiflora* Otth., *S. villosa* Forssk. and *S. vivianii* Steud. Smooth seeds were also reported in *S. littorea* Brot. [27] and *S. baccifera* (L.) Durande [28]. Papillose seeds have the lowest values of circularity and solidity both in lateral and dorsal views and are characterized by large tubercles. They correspond to seeds in *Silene* subg. *Behenantha* sect. *Physolychnis* (*S. laciniata* Cav. and *S. magellanica* (Desr.) Bocquet), to *S.* subg. *Behenantha* sect. *Behenantha* (*S. holzmannii* Heldr. *ex* Boiss.) and *S.* subg. *Silene* sect. *Silene* (*S. perlmanii* W.L. Wagner, D.R. Herbst and Sohmer). Rugose seeds have rounded tubercles, while those of echinate seeds are more acute, corresponding to lower and higher curvature values, respectively [27,28,29,30].

In addition to their utility in taxonomy, tubercle shape and distribution are of interest from a developmental perspective. Tubercle shape may be genetically determined, but it might also be subject to environmental regulation. Both the formation of tubercles as a developmental genetic process and their description in taxonomy require accurate quantification of tubercle size and shape. In recent work, we have presented and applied methods for the quantitative analysis of tubercle shape based on measurements of distances (tubercle width, height, and slope) and curvatures (maximum, average, and maximum to average curvature ratio) for 31 *Silene* species belonging to *S.* subg. *Silene* and *S.* subg. *Behenantha* [29,30]. The analysis, based on tubercle dimensions and curvature, revealed lower curvature values in species of *S.* subg. *Silene* in comparison with *S.* subg. *Behenantha*, and all but three species of *S.* subg. *Behenantha* grouping together in a dendrogram, corresponding to higher curvature values [30].

Nevertheless, an open question concerns whether tubercle type is conserved across populations of a given species, and to what extent tubercle curvature values are conserved. This aspect was addressed by Tabaripur et al. [31], who reported infraspecific variation in *S. odontopetala* Fenzl and also Wyatt [8] for *Arenaria uniflora* (Walter) Muhl. 

The objective of this work was to describe the tubercle structure quantitatively, measuring the tubercle height and width, maximum and average curvature values, and comparing them in diverse populations of eight species of *Silene*. The results showed a large variation in tubercle measurements among populations of most species, together with the identification of tubercle types characterized by high curvature values. These results are discussed in view of the application of these new techniques to the study of taxonomy and biodiversity.

## 2. Results

### 2.1. General Morphological Measurements

#### 2.1.1. Differences between Subgenera Based on Seed Measurements

Table 1 presents the results of the comparison of general morphological measurements of the lateral view of seeds between *Silene* subg. *Behenantha* and *Silene* subg. *Silene*. Differences between subgenera were found for all the measurements, with higher values for area, perimeter, length, width, roundness and solidity in *S.* subg. *Behenantha* and higher values of circularity and aspect ratio in *S.* subg. *Silene*. 

#### 2.1.2. Differences between Species

Differences between species were found in all general measurements (Table 2). *S. inaperta* had the lowest values of area, perimeter, length, and width. *S. nocturna* and *S. otites* had second lowest values of all these measurements, followed by *S. conica*. The highest area and width values corresponded to *S. diclinis*, followed by *S. vulgaris*, *S. dioica*, and *S. latifolia*. Perimeter values were highest in *S. vulgaris* and *S. diclinis*, followed by *S. dioica*, and *S. latifolia.* Circularity was lowest in *S. dioica* and *S. vulgaris* while the upper values corresponded to *S. conica* and *S. inaperta*. Aspect ratio was lowest in *S. conica*, followed by *S. dioica*. Highest values of aspect ratio were present in *S. diclinis*, *S. nocturna* and *S. otites.* The remaining species had intermediate values. According to solidity, the species were structured in five groups. The lowest values were found in *S. dioica*, followed by *S. vulgaris*, then by *S. nocturna*, with higher values in a group formed by *S. diclinis*, *S. latifolia* and *S. inaperta*. Highest solidity values corresponded to *S. conica*. The values of *S. otites* were between those of *S. vulgaris* and *S. nocturna*. 

#### 2.1.3. Differences between Populations of Each Species

Differences between populations for the general measurements related to seed morphology were found in all species. Tables containing the comparison of values between populations for each species are shown in Appendix B, Table A1, Table A2, Table A3, Table A4, Table A5, Table A6, Table A7 and Table A8.

For *S. conica*, area values were highest in the population of Alicante (AJ300) and lowest in the seeds of the populations of Ossa and JL3. The population of Ossa had the lowest values of circularity and the highest values corresponded to AJ300 and AJ76253. The highest solidity corresponded to AJ76253 and JL3, with less pronounced tubercles, while the lowest solidity corresponded to the Ossa population.

There were differences in all measurements between the populations of *S. diclinis*. Area was significantly lower for the Ranes population, while the highest values were observed for the population JL06011. Circularity values were highest in JL06011 and lowest in Ranes. The highest solidity was observed in JL06011 and JL5 and the lowest in JL04003.

Between the two populations of *S. dioica*, there were differences for all measurements except aspect ratio and roundness, with all size measurements higher in seeds of JL6, and lower values of circularity and solidity in JL6 than Pl02.

In *S. latifolia* there were differences in size measurements with the highest area in Xagó and the lowest in Pl04. Circularity was highest in AJ312. Solidity was highest in Xagó and lowest in JBU0373 and JL10.

Differences in area were found for *S. vulgaris* with the lowest values for all size measurements in JBUV2654, and the highest in AJ311 and Salamanca. Values of circularity were lower in the Salamanca population, and there were two groups of solidity with lower values in Salamanca and JBUV2654.

In *S. inaperta*, the seeds in populations of Alicante were smaller than the population stored at IBP, Brno. Values of area, perimeter, length and width were the lowest in AJ270. Circularity, in contrast, was the highest in JL1 and lowest in AJ335. Solidity values were different between AJ270 and AJ335.

In *S. nocturna*, there were differences between populations for all measurements except aspect ratio and roundness. 

Among the populations of *S. otites*, the seeds of JL192 were the largest with higher values of area, perimeter, length and width; those of JL14 had smallest values, and JL06009, intermediate. In contrast, circularity values were lower in JL192 than in the other two populations. 

### 2.2. Tubercle Measurements

#### 2.2.1. Differences between Subgenera

The mean values of tubercle width (W), height (H), and slope were higher for *S.* subg. *Behenantha*, while maximum curvature, and mean curvature values were higher in *S.* subg. *Silene*. There were no differences between subgenera in the values of the ratio maximum to mean curvature (Table 3).

#### 2.2.2. Differences between Species

There were differences between species for all the tubercle measurements (Table 4). Tubercle width had a maximum value in *S. diclinis*, followed by a group formed by *S. conica*, *S. latifolia*, *S. vulgaris* and *S. otites*. *S. inaperta* had the lowest values of tubercle width. Tubercle height was highest in *S. vulgaris*, followed by a group formed by *S. dioica* and *S. diclinis. S. inaperta* had the lowest values of height. Related to slope, two groups were differentiated with higher values: *S. vulgaris* and *S. dioica* in one group, followed by *S. latifolia* and *S. diclinis*. Maximum curvature values resulted in two groups of higher values and one with lower values: higher values were obtained for the seeds of *S. dioica*, followed by *S. vulgaris* and *S. nocturna*; *S. diclinis* presented the lowest values of maximum curvature. Mean curvature values grouped the seeds into five groups with *S. dioica* having the highest scores, followed by *S. inaperta* and *S. nocturna*, a third group formed by *S. latifolia* and *S. vulgaris*, a fourth group formed by *S. otites*, and the lowest values in *S. conica* and *S. diclinis*. Lowest values of the ratio max to mean curvature were found in *S. inaperta* and *S. latifolia.*

#### 2.2.3. Differences between Populations for Each Species

Significant differences between populations for the measurements related to tubercle size and shape (curvature) were found for all species, with the only exception being the two populations of *S. dioica*. The corresponding data are shown in Appendix B, Table A9, Table A10, Table A11, Table A12, Table A13, Table A14, Table A15 and Table A16. Images of seeds with the curvature analysis of representative tubercles are shown in Figure 1, Figure 2, Figure 3, Figure 4, Figure 5, Figure 6, Figure 7, Figure 8, Figure 9 and Figure 10.

For *S. conica* (Table A9; Figure 1 and Figure 2), the mean tubercle width was comprised between the lowest value of 50.0 μm found in the population of Ossa and the highest corresponding to AJ76253 and AJ300, respectively. The mean width value in JL3 was intermediate. Tubercle height and slope formed three groups with the lowest values in JL3, intermediate in AJ300 and AJ76253, and highest in Ossa. Differences were also found for the maximum, and mean curvature values and max to mean curvature ratio. Maximum curvature values were higher in Ossa than in the other three populations (compare Figure 1 and Figure 2).

For *S. diclinis* (Table A10; Figure 3 and Figure 4), the tubercle width had lower values in JL5, JL04003 and Ranes and higher in JL06011. Tubercle height formed two groups with the lowest values in JL5 and the highest in the remaining three populations. Tubercle slope and maximum curvature were higher in the group formed by JL04003 and Ranes, and lower in JL06011 and JL5 (compare Figure 3 and Figure 4). The max to mean curvature ratio resulted in four groups, one for each population. 

There were no differences in tubercle measurements between the two populations of *S. dioica* (Table A11).

For *S. latifolia* (Table A12; Figure 5 and Figure 6) there were differences for all measurements. Tubercle width had the lowest value in the group formed by Pl04 and JBUV1444, and the highest in AJ312. Tubercle height formed three groups with the lowest values in Xagó, the highest in the group formed by JBUV0373 and JL10, and intermediate for the remaining three populations. Tubercle slope was highest in JL10 and JBUV0373, followed by the group formed by JBUV1444 and Pl04, and was lowest in the population from Xagó. Differences were also found for the maximum and mean curvature values and max to mean curvature ratio. The maximum curvature values were higher in the group formed by JBUV1444, JL10 and Pl04, and lower in the group formed by AJ312 and Xagó (see Figure 5 for the curvature analysis in the representative tubercles of population JL10 and Figure 6 for the curvature analysis in the representative tubercles of population Xagó). The max to mean curvature ratio was lower in Xagó than in the other populations, showing lower variation in curvature along the curve, related to the similarity between the tubercle silhouette and an arc of circumference.

Related to *S. vulgaris* (Table A13; Figure 7 and Figure 8), the results for width, maximum and mean curvature formed two groups, one formed by AJ311 and the other by the remaining three populations. The seeds of AJ311 had higher width and lower curvature values than the group formed by the other three populations. Tubercle slope was highest in JBUV2654, lowest in AJ311 and intermediate in the other two populations. 

For *S. inaperta* (Table A14) there were differences in all measurements between populations. Tubercle width was smaller in the JL1 than in the other two populations. Tubercle slope was higher in a group formed by AJ270 and JL1, lower in AJ335. Differences were also found for the maximum and mean curvature values and the max to mean curvature ratio. The maximum curvature formed three groups, with the highest values in JL1, the lowest in AJ335, and intermediate in AJ270. The mean curvature was higher in JL1 and lower in the group formed by the other two populations.

Related to *S. nocturna* (Table A15; Figure 9 and Figure 10), tubercle height was lower in JL2062 than in the group formed by the other four populations. The maximum curvature was higher in AJ287 than in the group formed by the other populations, and the mean curvature was lower in AJ47439 than in the group formed by the other populations.

For the populations of *S. otites*, related to tubercle width and height, three groups were defined, each formed by a single population. In these cases, values were highest in JL14, intermediate in JL06009, and lowest in JL192. There were also differences in the curvature values with the maximum and mean curvature values superior in the JL14, and lower values in the group formed by JL06009 and JL192.

## 3. Discussion

With more than 800 species distributed mainly in temperate regions of the Northern Hemisphere, *Silene* is one of the largest genera in the Caryophyllaceae. The taxonomy of *Silene*, long based on morphological characters and more recently on DNA sequence data, has been a matter of debate for decades [32,33,34,35]. Three subgenera, named as *S.* subg. *Lychnis* (L.) Greuter, *S.* subg. *Behenantha*, and *S.* subg. *Silene*, are today accepted based on DNA sequence analysis [34]. The *S.* subg. *Lychnis* contains four sections with about 20 species native from Europe, Asia, N and E Africa. The *S.* subg. *Behenantha* includes 18 sections with species previously classified in *Cucubalus* L., *Gastrolychnis* (Fenzl) Rchb., *Melandrium* Röhl. and *Pleconax* Raf., while *S.* subg. *Silene* contains 11 sections, including *S.* sect. *Auriculatae* (Boiss.) Schischk., *S.* sect. *Rigidulae* [36], *S.* sect. *Silene* [36], *S.* sect. *Sclerocalycinae* (Boiss.) Schischk. [36], *S.* sect. *Siphonomorpha* Otth s.l. [37] and several unassigned groups including the Hawaiian endemics [36]. 

Given this complex panorama, it is challenging to attribute morphological properties unequivocally to subgenera and sections. However, traditional seed characters can be quantified and associated with certain taxonomic groups. For instance, seeds described as dorso plana and dorso canaliculata by Boissier and Rohrbach [32,33] correspond to convex and non-convex seeds in their dorsal views and they are related to species and sections in *S.* subg. *Behenantha* and *S.* subg. *Silene*, respectively [38]. Geometric models have allowed for the quantification of seed shape in *Silene* species, revealing that more convex lateral models are associated with species belonging to *S.* sect. *Melandrium* in *S.* subg. *Behenantha*, while non-convex models fit better to seeds in *S.* subg. *Silene* [39]. Furthermore, seed surface characteristics may vary among subgenera, with smooth and rugose seeds more common in *S.* subg. *Silene*, while echinate-type seeds are more frequent in *S.* subg. *Behenantha* [28]. Differences in the seed surface due to the presence of tubercles can also be found between different genera. Members of *Heliosperma* (Rchb.) Rchb, which were ascribed in the past to *Silene*, and can be told apart by the crest of long tubercles on the seeds [40].

Quantitative methods have recently been applied to measure tubercle length, width, and curvature in seeds of 31 species, with 12 belonging to *S.* subg. *Behenantha* and 19 to *S.* subg. *Silene* [30]. Results from this work pointed to an association between higher values of tubercle height and slope and higher values of maximum and average curvature and maximum to mean curvature ratio in *S.* subg. *Behenantha* [30]. Our results presented here support higher values of tubercle height and slope in *Silene* subg. *Behenantha*, as stated [30]. Nevertheless, the analysis of multiple populations of diverse origins does not support higher values of curvature, in general, for this subgenus. The obtained data remarkably varied among populations, and the presence of higher values was not a general rule for all of the studied species and their populations. These different results can be explained by the origin of the seeds, since these new data were obtained based on a high proportion of Spanish populations (see later). 

Seed shape diversity and stability in populations are aspects that have received little attention in the literature. Prentice and Runyeon [41] and Prentice [42,43] studied the distribution of morphological types and diversity of tubercles in some species of the genus. Prentice proposed an association between geographical longitude and tubercle shape and length in European populations of *S. latifolia* and *S. dioica* [43]. This study revealed that tubercles of the tall type predominate in *S. latifolia* seeds from the European cold-winter region (north and east of the 0 °C isotherm), while low tubercle type was more abundant in the warm-winter region [43]. Accordingly, among the six populations of *S. latifolia* used in this study, the two populations (AJ312 and Xagó) from the Iberian Peninsula, which correspond to the second described region by Prentice [43], had the lowest values of maximum and mean curvature. Thus, the value of 55 mm^−1^ reported here for maximum curvature for the six *S. latifolia* populations contrasts with 97.8 μm^−1^ obtained in a previous report for *S. latifolia* (code JBUV1444) from Germany [29]. In addition to considering latitude, altitude may be also significant in this respect. Our data pointed out that the populations of *S. latifolia* with rounded tubercles (low curvature values) come from Pego (Alicante) and Xagó (Asturias), and both localities are quite close to sea level. To clarify and support this relationship between altitude and the presence of rounded tubercles within the same species, further studies are needed.

Maximum curvature values reported in earlier work for *S. inaperta*, *S. nocturna*, and *S. otites* were 54.1, 30.3, and 75.6 mm^−1^, respectively, while the values reported here for these species are 61, 83, and 52 mm^−1^, respectively. The highest differences between the obtained measures were found for *S. nocturna*. These different results can be partially attributed to an elevated maximum curvature found here in the population AJ287 of 108 μm^−1^. The increased maximum curvature of this population AJ287 of *S. nocturna* is due to the existence of several tubercles that are characterized by acute protuberances in their central part. This peculiar morphology has been described as mammillae, and hence, the seeds with this feature are named as mamillated [44] or umbonated [8]. Mammillae are frequent in seeds of some sections such as *S.* sect. *Siphonomorpha* but absent in other sections [44]. We reported them before in *S. gigantea* (L.) L. and *S. spinescens* Sm., both species of S. subg. *Silene* sect. *Siphonomorpha*, and in *S.*
*squamigera* subsp. *vesiculifera* (J.Gay ex Boiss.) Coode and Cullen, included in *S.* subg. *Silene* sect. *Lasioclycinae* [30].

Differences in curvature values among populations of the same species have been detected for most of the species studied, and the cases of *S. latifolia* and *S. nocturna* have been discussed in the preceding paragraphs. Among the remaining species, of particular interest the case of *S. conica*. Among the populations used for this species, higher curvature values were found for the seeds from Ossa de Montiel, close to Laguna Redondilla (Parque Natural Lagunas de Ruidera) at an altitude of approximately 900 m. In this context, it may be interesting to study the variation in tubercle size and shape at different altitudes.

The morphological characteristics of *Silene* seeds make this genus a unique model for genetic analysis of development. The overall seed shape can be described and quantified by comparison with geometric models, and peculiarities of the seed surface, shape, and distribution of tubercles can also be analysed. Quantitative analysis of the seed surface is also a promising tool for taxonomy in other species of the *Caryophyllaceae* as well as members of other families in the *Caryophyllales*. The relief of the periclinal cell wall presents interesting variations in the *Portulacaceae*, and the morphological types described by Ocampo [45], such as flat, convex, low-convex, high-convex, and par-convex, could be characterized by quantitative measurements of tubercle height, width, and curvature. Quantitative analysis of seed surface tubercles may be a tool to identify genetic and environmental factors involved in their regulation.

## 4. Materials and Methods

### 4.1. Species and Populations of Silene Analysed

Eight species of *Silene* were selected, belonging to *S.* subg. *Behenantha* (*S. conica*, *S.*
*diclinis*, *S. dioica*, *S. latifolia* and *S. vulgaris*) and to *S.* subg. *Silene* (*S. inaperta*, *S. nocturna* and *S. otites*). Up to six populations per species and 20 seeds per population were studied. Details of the studied populations are listed in Table 5.

### 4.2. Seed Images

For the analysis of individual tubercles, photographs were taken with a Nikon Stereomicroscope Model SMZ1500 (Nikon, Tokyo, Japan), equipped with a camera Nikon DS-Fi1 of 5.24 megapixels (Nikon, Tokyo, Japan). Photographs were stored as JPG images of 2560 × 1920 with 300 ppp.

### 4.3. Seed Morphological Measurements

Seed area (A), perimeter (P), length (L), width (W), aspect ratio (AR), circularity (C), roundness (R) and solidity (S), were determined with Image J [46] on images of 20 seeds per population (620 seeds in total). The images are available at Zenodo (See Appendix A).

### 4.4. Tubercle Measurements

Tubercle measurements and curvature analyses were applied to seeds of 31 populations from eight species of *Silene*. To obtain accurate and reproducible measurements, well-defined and regular shaped tubers were selected. Small, irregular, or densely compacted tubercles can make measurements of curvature, height, and width at the base difficult or confusing.

Measurements of width, height, slope and curvature (maximum and average curvature values) were taken for a total of 396 tubercles (268 corresponding to rugose seeds and 128 to echinate seeds). The number of seeds was between one and three for each species, as indicated in Table 2 and Table 3.

#### 4.4.1. Width, Height and Slope Measurements in the Tubercles

Seed images containing a ruler of 1 mm were opened in Image J. Two direct measurements were made for each of the tubercles indicated in Figure 4: tubercle width at the base (W) and tubercle height (H). The slope of each tubercle (S) was calculated as S = (W/H) × 200.

#### 4.4.2. Curvature Measurements

Maximum absolute values and average curvatures were determined for individual tubercles of each species according to established procedures [47,48,49,50] (See Appendix A). In the curvature measurements for individual tubercles, the points were taken automatically with the Analyze line graph function from Image J. Among 6 and 22 tubercles of representative seeds were analysed for each species. Individual images in JPEG format were kept of each tubercle and vertically oriented. Images were opened in Image J and converted to 8-bit, threshold values were adjusted, and the image was analysed. The curve corresponding to the tubercle was selected from the outlines, and a new threshold was defined, before the corresponding line graph was analysed, to obtain the x,y coordinates. The coordinates were the basis of obtaining the Bézier curve and the corresponding curvature values according to published protocols [47,48,49,50,51]. Curvature was given in mm^−1^; thus, a curvature of 20 corresponds to a circumference of 50 microns (1/50 × 1000) and a curvature of 100 to a circumference of 10 microns.

### 4.5. Statistical Analysis

The mean, minimum, and maximum values and the standard deviation were obtained for all of the measurements indicated above (A, P, L, W, AR, C, R and S) as well as curvature measurements. Statistics were calculated with IBM SPSS statistics v28 (SPSS 2021) and R software v. 4.1.2 [51]. As some of the populations did not follow a normal distribution, non-parametric tests were applied for the comparison of populations. Kruskal–Wallis tests were used in the cases involving three or more populations followed by stepwise stepdown comparisons by the ad hoc procedure developed by Campbell and Skillings [52]; *p* values inferior to 0.05 were considered significant. The coefficient of variation was calculated as CVtrait = standard deviationtrait/meantrait × 100 [53].

## 5. Conclusions

Seed shape and size comparisons among different populations of eight species of *Silene* indicated a high degree of variation among populations. Quantitative analysis of seed tubercles also showed variations in size (tubercle height and width) and shape (curvature values). Notable differences in tubercle curvature values were found among the populations of *S. diclinis*, *S. conica*, and *S. latifolia*, and to a minor extent in *S. inaperta*, *S. nocturna*, *S. otites* and *S. vulgaris*. Among the tubercles analysed in *S. nocturna*, some of them belong to the type that was termed as mammillated or umbonated. These are characterized by high values of maximum curvature. The protocols developed here for the quantification of tubercle size and shape may be useful for investigating the morphological variation in the seeds of many species in the Caryophyllales as well as to investigate the genetic developmental changes in seeds of *Silene*.

## Figures and Tables

**Figure 1 plants-13-01416-f001:**
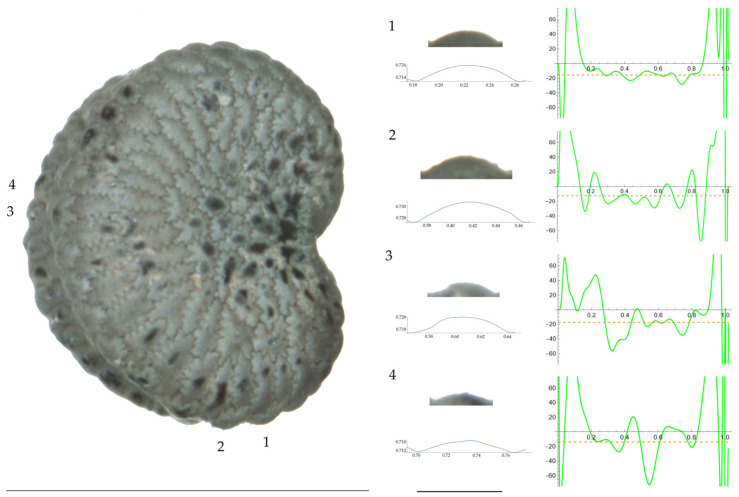
**Left**: A seed of *S. conica* from the AJ300 population with four tubercles labelled. Bar on the left represents 1 mm. **Right**: individual view of the tubercles, Bézier curves (below each tubercle) and curvature plots representing the variation in curvature along the Bézier curve. The discontinuous line in the curvature plot represents the value of the mean curvature. Bar on the right side below the tubercle images represents 100 μm.

**Figure 2 plants-13-01416-f002:**
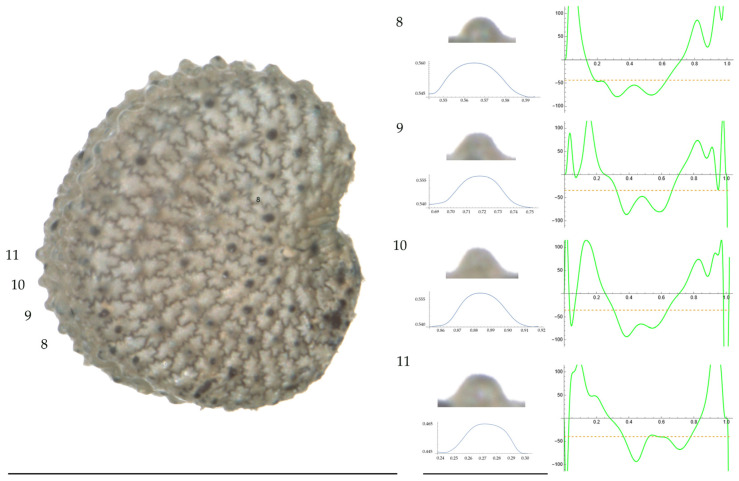
**Left**: A seed of *S. conica* from the Ossa population with four tubercles labelled. Bar on the left represents 1 mm. **Right**: individual view of the tubercles, Bézier curves (below each tubercle) and curvature plots representing the variation in curvature along the Bézier curve. The discontinuous line in the curvature plot represents the value of mean curvature. Bar on the right side below the tubercle images represents 100 μm.

**Figure 3 plants-13-01416-f003:**
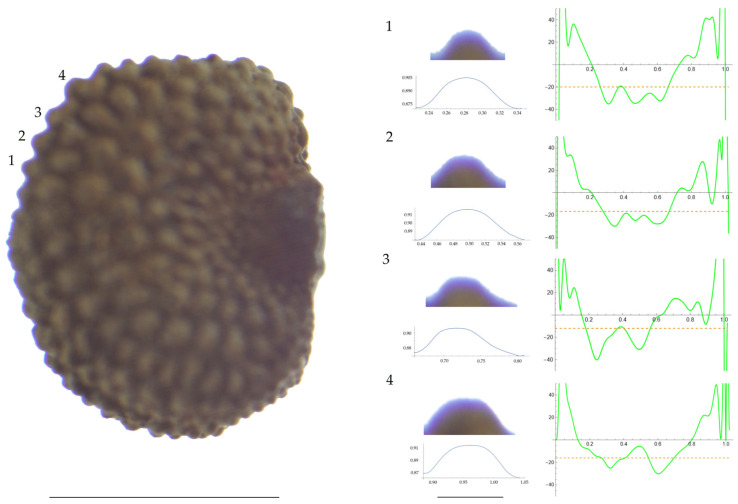
**Left**: A seed of *S. diclinis* from the JL06011 population with four tubercles labelled. Bar on the left represents 1 mm. **Right**: individual view of the tubercles, Bézier curves (below each tubercle) and curvature plots representing the variation in curvature along the Bézier curve. The discontinuous line in the curvature plot represents the value of mean curvature. Bar on the right side below the tubercle images represents 100 μm.

**Figure 4 plants-13-01416-f004:**
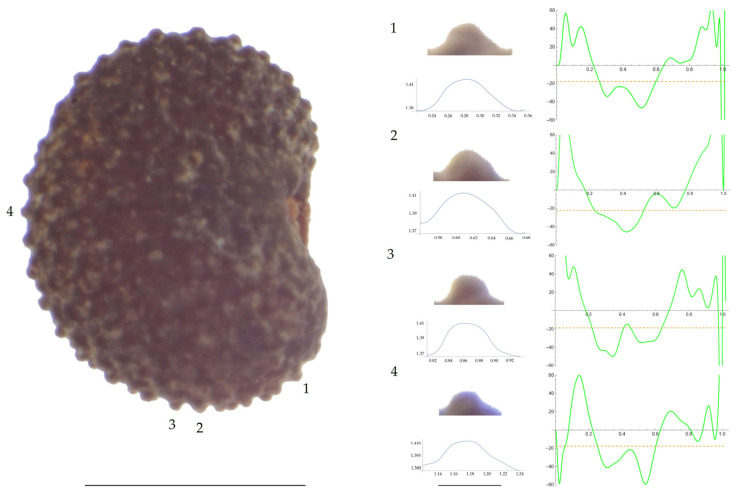
**Left**: A seed of *S. diclinis* from the Ranes population with four tubercles labelled. Bar on the left represents 1 mm. **Right**: individual view of the tubercles, Bézier curves (below each tubercle) and curvature plots representing the variation in curvature along the Bézier curve. The discontinuous line in the curvature plot represents the value of mean curvature. Bar on the right side below the tubercle images represents 100 μm.

**Figure 5 plants-13-01416-f005:**
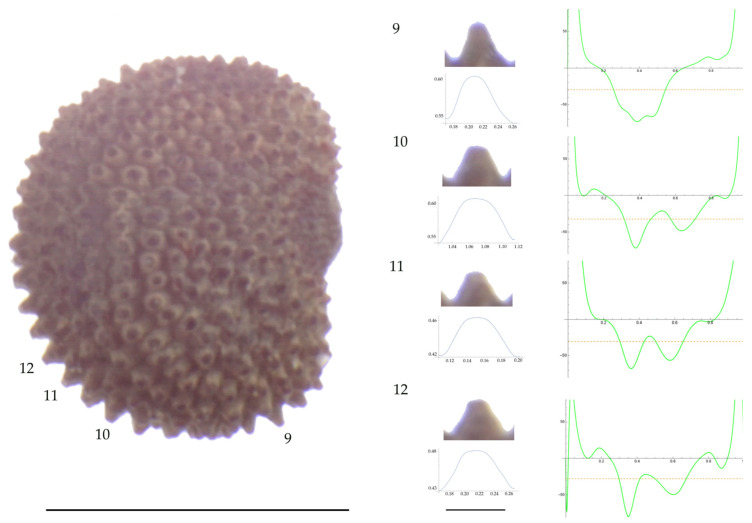
**Left**: A seed of *S. latifolia* from the JL10 population with four tubercles labelled. Bar on the left represents 1 mm. **Right**: individual view of the tubercles, Bézier curves (below each tubercle) and curvature plots representing the variation in curvature along the Bézier curve. The discontinuous line in the curvature plot represents the value of mean curvature. Bar on the right side below the tubercle images represents 100 μm.

**Figure 6 plants-13-01416-f006:**
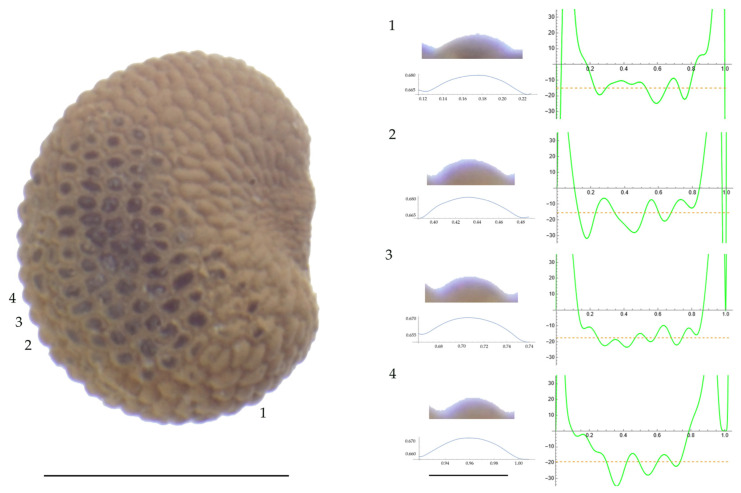
**Left**: A seed of *S. latifolia* from the Xagó population with four tubercles labelled. Bar on the left represents 1 mm. **Right**: individual view of the tubercles, Bézier curves (below each tubercle) and curvature plots representing the variation in curvature along the Bézier curve. The discontinuous line in the curvature plot represents the value of mean curvature. Bar on the right side below the tubercle images represents 100 μm.

**Figure 7 plants-13-01416-f007:**
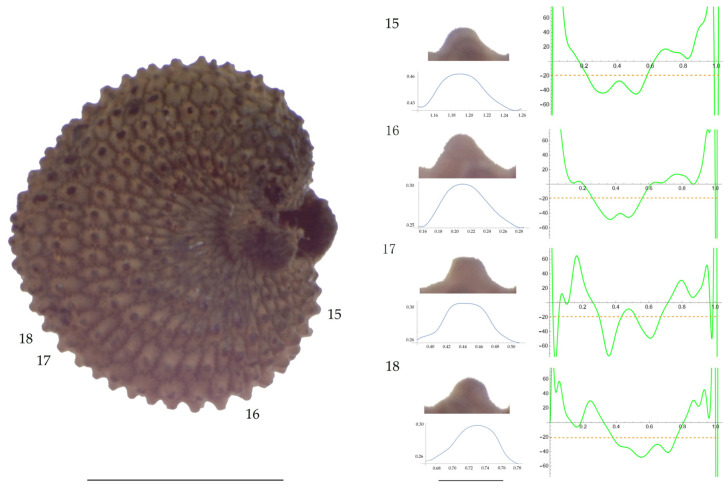
**Left**: A seed of *S. vulgaris* from the AJ311 population with four tubercles labelled. Bar on the left represents 1 mm. **Right**: individual view of the tubercles, Bézier curves (below each tubercle) and curvature plots representing the variation in curvature along the Bézier curve. The discontinuous line in the curvature plot represents the value of mean curvature. Bar on the right side below the tubercle images represents 100 μm.

**Figure 8 plants-13-01416-f008:**
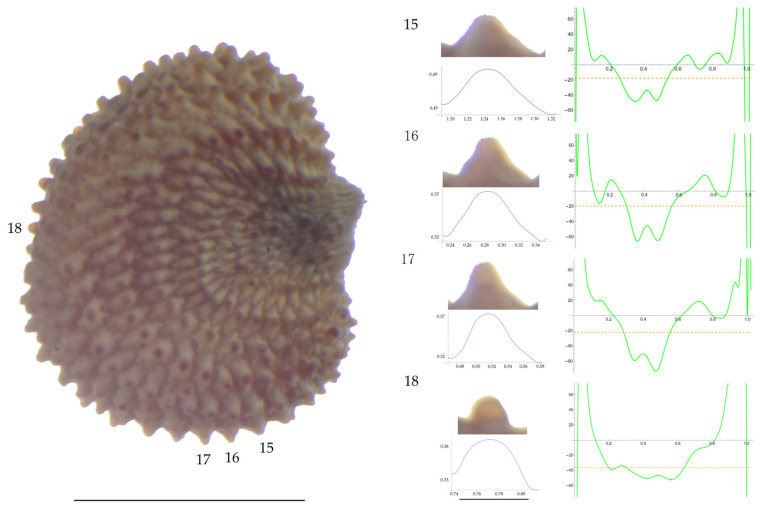
**Left**: A seed of *S. vulgaris* from the Salamanca population with four tubercles labelled. Bar on the left represents 1 mm. **Right**: individual view of the tubercles, Bézier curves (below each tubercle) and curvature plots representing the variation in curvature along the Bézier curve. The discontinuous line in the curvature plot represents the value of mean curvature. Bar on the right side below the tubercle images represents 100 μm.

**Figure 9 plants-13-01416-f009:**
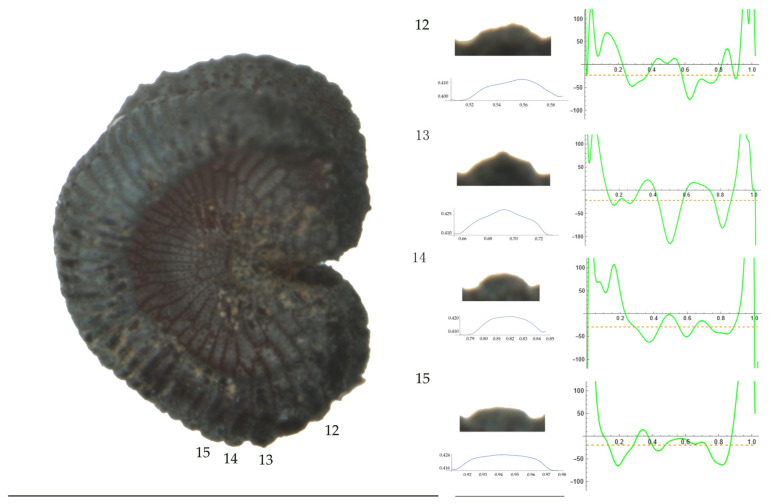
**Left**: A seed of *S. nocturna* from the AJ47439 population with four tubercles labelled. Bar on the left represents 1 mm. **Right**: individual view of the tubercles, Bézier curves (below each tubercle) and curvature plots representing the variation in curvature along the Bézier curve. The discontinuous line in the curvature plot represents the value of mean curvature. Bar on the right side below the tubercle images represents 100 μm.

**Figure 10 plants-13-01416-f010:**
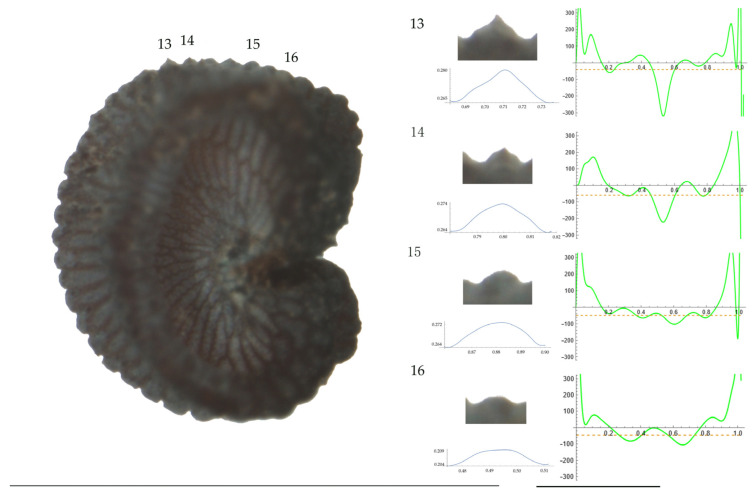
**Left**: A seed of *S. nocturna* from the AJ287 population with four tubercles labelled. Bar on the left represents 1 mm. **Right**: individual view of the tubercles, Bézier curves (below each tubercle) and curvature plots representing the variation in curvature along the Bézier curve. The discontinuous line in the curvature plot represents the value of mean curvature. Bar on the right side below the tubercle images represents 100 μm.

**Table 1 plants-13-01416-t001:** Results of Kruskal–Wallis test for the comparison of area (A), perimeter (P), length (L), width (W), circularity (C), aspect ratio (AR), roundness (R) and solidity (S) between species of *Silene* subg. *Behenantha* and *S.* subg. *Silene*. Mean values of P, L and W are given in mm, A in mm^2^. N is the number of seeds analysed. Coefficients of variation are given between parentheses. The minimum and maximum values are given below each mean value. Different superscript letters indicate significant differences between files in the same column (*p* < 0.05).

	N	A	P	L	W	C	AR	R	S
*Behenantha*	400	1.31 ^b^ (36.9) 0.49–2.52	5.31 ^b^ (27.2) 2.88–9.46	1.39 ^b^ (21.8) 0.84–1.99	1.15 ^b^ (20.0) 0.71–1.64	0.59 ^a^ (20.5) 0.26–0.81	1.20 ^a^ (5.3) 1.01–1.44	0.83 ^b^ (5.2) 0.69–0.99	0.963 ^b^ (1.2) 0.926–0.985
*Silene*	240	0.50 ^a^ (29.8) 0.30–1.12	3.01 ^a^ (16.0) 2.37–4.80	0.88 ^a^ (14.3) 0.68–1.31	0.71 ^a^ (13.5) 0.56–1.08	0.69 ^b^ (8.1) 0.52–0.79	1.24 ^b^ (5.3) 1.07–1.50	0.81 ^a^ (5.2) 0.67–0.94	0.959 ^a^ (0.9) 0.935–0.979

**Table 2 plants-13-01416-t002:** Seed morphological measurements. Results of Kruskal–Wallis test for the comparison of area (A), perimeter (P), length (L), width (W), circularity (C), aspect ratio (AR), roundness (R) and solidity (S) between eight species of *Silene*. Values of P, L and W are given in mm, A in mm^2^. N is the number of seeds analysed. Coefficients of variation are given between parentheses. The minimum and maximum values are given below each mean value. Different superscripts indicate significant differences between files in the same column (*p* < 0.05). The five upper rows correspond to *Silene* subg. *Behenantha*, and the three lower rows to *S.* subg. *Silene*.

	N	A	P	L	W	C	AR	R	S
*S. conica*	80	0.59 ^c^ (8.6) 0.49–0.72	3.19 ^c^ (4.6) 2.88–3.57	0.93 ^c^ (5.0) 0.84–1.03	0.81 ^c^ (4.8) 0.71–0.89	0.74 ^e^ (7.4) 0.60–0.81	1.15 ^a^ (4.7) 1.01–1.37	0.87 ^e^ (4.6) 0.73–0.99	0.971 ^e^ (0.8) 0.948–0.983
*S. diclinis*	80	1.92 ^g^ (11.5) 1.39–2.52	6.34 ^f^ (8.6) 4.98–7.99	1.74 ^f^ (5.9) 1.49–1.99	1.40 ^g^ (6.0) 1.18–1.61	0.60 ^b^ (12.5) 0.41–0.73	1.25 ^e^ (3.2) 1.16–1.34	0.80 ^a^ (3.2) 0.75–0.87	0.966 ^d^ (0.9) 0.940–0.980
*S. dioica*	40	1.25 ^e^ (15.4) 0.92–1.57	5.93 ^e^ (16.1) 4.56–8.07	1.37 ^d^ (8.0) 1.16–1.55	1.16 ^e^ (7.9) 0.98–1.30	0.46 ^a^ (19.7) 0.28–0.61	1.18 ^b^ (3.5) 1.11–1.30	0.85 ^d^ (3.4) 0.77–0.90	0.949 ^a^ (1.1) 0.928–0.966
*S. latifolia*	120	1.16 ^d^ (16.4) 0.80–1.49	4.95 ^d^ (12.1) 4.11–6.85	1.34 ^d^ (8.9) 1.12–1.63	1.10 ^d^ (8.4) 0.90–1.27	0.60 ^b^ (16.1) 0.34–0.80	1.22 ^d^ (4.8) 1.08–1.41	0.82 ^b^ (4.7) 0.71–0.93	0.967 ^d^ (1.0) 0.940–0.985
*S. vulgaris*	80	1.67 ^f^ (25.0) 0.86–2.34	6.63 ^f^ (18.0) 4.43–9.46	1.58 ^e^ (13.4) 1.12–1.88	1.32 ^f^ (13.1) 0.98–1.64	0.48 ^a^ (16.1) 0.26–0.61	1.20 ^bc^ (5.5) 1.09–1.44	0.84 ^cd^(5.3) 0.69–0.92	0.953 ^b^ (0.7) 0.926–0.971
*S. inaperta*	60	0.43 ^a^ (10.4) 0.32–0.59	2.71 ^a^ (5.8) 2.37–3.15	0.81 ^a^ (5.5) 0.69–0.95	0.67 ^a^ (5.4) 0.59–0.79	0.73 ^e^ (5.7) 0.57–0.79	1.21 ^cd^ (3.7) 1.13–1.31	0.83 ^bc^(3.6) 0.76–0.88	0.967 ^d^ (0.4) 0.956–0.977
*S. nocturna*	120	0.49 ^b^ (18.7) 0.30–0.73	3.04 ^b^ (10.2) 2.37–3.93	0.88 ^b^ (10.0) 0.68–1.13	0.70 ^b^ (9.4) 0.56–0.85	0.67 ^c^ (6.8) 0.52–0.79	1.25 ^e^ (5.4) 1.07–1.50	0.80 ^a^ (5.3) 0.67–0.94	0.957 ^c^ (0.8) 0.935–0.975
*S. otites*	60	0.60 ^b^ (39.5) 0.36–1.12	3.25 ^b^ (23.2) 2.42–4.80	0.96 ^bc^ (19.5) 0.75–1.31	0.76 ^b^ (19.5) 0.59–1.08	0.70 ^d^ (8.5) 0.57–0.79	1.26 ^e^ (5.4) 1.09–1.43	0.79 ^a^ (5.4) 0.70–0.92	0.956 ^bc^ (0.9) 0.940–0.979

**Table 3 plants-13-01416-t003:** Results of Kruskal–Wallis test for the comparison of tubercle width (W), height (H), maximum curvature (C max), mean curvature (C mean), and ratio (C max/C mean), between species belonging to *S.* subg. *Behenantha* and *S.* subg. *Silene*. N is the number of seeds analysed. Values of W and H are given in mm, C max and C mean in mm^−1^. Coefficients of variation are given between parentheses. The minimum and maximum values are given below each mean value. Different superscript letters indicate significant differences between files in the same column (*p* < 0.05).

Subgenera	N *	W	H	Slope	C Max	C Mean	Ratio
*Behenanta*	396	68.9 ^b^ (27.8) 35.2–125.9	34.6 ^b^ (44.0) 5.5–89.8	104.2 ^b^ (43.2) 20.6–196.4	61 ^a^ (53.2) 22–349	26 ^a^ (40.1) 8–60	2.45 ^a^ (38.8) 1.18–8.97
*Silene*	239	51.9 ^a^ (35.0) 25.0–109.4	11.1 ^a^ (46.9) 3.6–32.4	42.4 ^a^ (25.6) 21.8–82.3	69 ^b^ (52.1) 28–395	28 ^b^ (33.9) 9–67	2.54 ^a^ (43.5) 1.30–12.08

* Total number of seeds is slightly lower than in Table 1 because not all seeds were used for tubercle analysis.

**Table 4 plants-13-01416-t004:** Results of Kruskal–Wallis test for the comparison of tubercle measurements between species. Mean values and differences between species in width (W), height (H), slope (S), maximum curvature (C Max), mean curvature (C Mean) and maximum curvature to mean curvature ratio are shown. N is the number of tubercles analysed (between parentheses number of seeds). Values of W and H are given in μm, C max and C mean in mm^−1^. Coefficients of variation are given between parentheses. The minimum and maximum values are given below each mean value. Different superscript letters indicate significant differences between files in the same column (*p* < 0.05). The five upper rows correspond to *Silene* subg. *Behenantha*, and the three lower rows to *S.* subg. *Silene*.

Species	N	W	H	Slope	C Max	C Mean	Ratio
*S. conica*	76 (12)	66.2 ^d^ (21.9) 35.2–114.8	14.1 ^c^ (35.1) 5.5–28.1	44.7 ^ab^ (46.1) 20.6–98.1	54 ^b^ (44.6) 28–114	29 ^a^ (45.2) 8–48	2.01 ^cd^ (44.7) 1.4–9.0
*S. diclinis*	80 (18)	91.4 ^e^ (18.4) 60.3–125.9	43.7 ^e^ (29.1) 23.5–89.8	96.7 ^d^ (25.6) 60.2–175.9	40 ^a^ (28.8) 23–85	27 ^a^ (25.8) 9–30	1.53 ^b^ (31.7) 1.4–5.7
*S. dioica*	40 (12)	50.6 ^c^ (16.5) 36.9–77.0	38.8 ^e^ (16.9) 26.1–49.5	154.2 ^e^ (13.7) 118.7–196.4	113 ^e^ (51.0) 60–349	60 ^e^ (20.8) 23–60	1.88 ^cd^ (42.4) 1.6–7.4
*S. latifolia*	120 (21)	64.7 ^d^ (25.7) 39.8–123.8	32.0 ^d^ (35.5) 12.8–60.4	102.8 ^d^ (35.2) 43.6–178.1	55 ^bc^ (36.3) 22–102	40 ^c^ (30.1) 11–44	1.38 ^a^ (27.3) 1.2–5.0
*S. vulgaris*	80 (5)	64.5 ^d^ (19.8) 39.0–93.2	46.6 ^f^ (21.7) 21.7–76.8	145.3 ^e^ (15.0) 96.0–186.2	74 ^d^ (26.8) 39–125	38 ^c^ (39.7) 14–47	2.29 ^d^ (61.3) 1.4–5.4
*S. inaperta*	60 (4)	40.1 ^a^ (17.8) 25.0–56.4	7.3 ^a^ (27.8) 3.6–13.2	36.5 ^a^ (20.6) 21.8–53.7	61 ^c^ (27.8) 38–123	47 ^d^ (29.8) 17–67	1.33 ^a^ (22.2) 1.4–3.8
*S. nocturna*	119 (15)	47.3 ^b^ (21.4) 29.6–84.3	10.6 ^b^ (28.9) 4.5–20.1	45.0 ^c^ (22.9) 28.0–81.6	83 ^d^ (57.8) 33–395	45 ^d^ (28.2) 15–59	1.89 ^cd^ (50.4) 1.3–12.1
*S. otites*	60 (8)	72.6 ^d^ (29.4) 39.5–109.4	15.9 ^c^ (43.7) 6.0–32.4	43.1 ^bc^ (28.8) 26.9–82.3	52 ^b^ (36.3) 28–118	32 ^b^ (34.8) 9–41	1.70 ^c^ (33.5) 1.6–7.2

**Table 5 plants-13-01416-t005:** Populations and origin of the seeds used in the present work, with the indication of their corresponding codes. Taxonomic adscription of species to sections and subgenera according to Jafari et al. [34]. U means unknown origin.

Name of the Species (Section, Subgenus)	Population Code	Details of the Population and Origin of the Seeds
*Silene conica* L.	AJ300	Villena, Alicante (Spain)
(*Conoidea*, *Behenantha*)	AJ76253	Albarracín, Albacete
	Ossa	Ossa de Montiel. Laguna redondilla (Parque Natural Lagunas de Ruidera; 24/06/2023)
	JL3	Berlin (IBP collection, Brno)
*Silene diclinis* (Lag.) M. Laínz	JL04003	U. (IBP collection, Brno)
(*Melandrium*, *Behenantha*)	Ranes	Ermita de Santa Ana. La llosa de Ranes. (06/2023)
	JL5	Plá de Mora (IBP collection, Brno)
	06011	U. (IBP collection, Brno)
*Silene dioica* (L.) Clairv.	JL7	Tišnov (CzR) (IBP collection, Brno)
(*Melandrium*, *Behenantha*)	Pl02	U. (University of Lodz, Poland)
*Silene inaperta* L.	AJ270	Rambla Bateig, Elda, Alicante, Spain (2021)
(*Muscipula*, *Silene*)	AJ335	Rambla de Caprala, Petrer, Alicante, Spain (2021)
	JL7	U. (IBP collection, Brno)
*Silene latifolia* Poir.	AJ312	Pego, Alicante, Spain
(*Melandrium*, *Behenantha*)	JBUV373	Arcins, Marais (Jardin Botanique de Bordeaux)
	JBUV1444	Germany, Brandenburg, Landkreis Märkisch-Oderland, Petershagen (Humboldt Universität, Berlin)
	JL10	Panenská Rozsíčka (IBP collection, Brno)
	Pl04	Dubidze (University of Lodz, Poland)
	Xagó	Playa de Xagó (Asturias, Spain)
*Silene nocturna* L.	AJ287	Campo de cultivo, Villena, Alicante
(*Silene*, *Silene*)	AJ316	Sierra de Irta, Torre Badum, Peñíscola, Castellón
	AJ322	Forcall, camino a Villores sobre roca, bosque Quercus (Castellón)
	AJ47439	L’Abdet, Alicante, Spain
	JL206B3	U. (IBP collection, Brno)
*S. otites* (L.) Wibel	JL06009	U. (IBP collection, Brno)
(*Siphonomorpha*, *Silene*)	JL14	Rohatec (IBP collection, Brno)
	JL192	U. (IBP collection, Brno)
*S. vulgaris* (Moench) Garcke	AJ311	Elda, Alicante, Spain
(*Behenantha*, *Behenantha*)	JBUV216	Mt. Naszály near Vác, rocky grassland on dolomite at the ridge of Szarvashegy Botany Hung. Acad. of Sciences
	JBUV2654	Persenbeug, Niederösterreich, Austria (Meise Botanic Garden)
	Salamanca	c/Sectores 37b y 67a. Salamanca.

## Data Availability

The original contributions presented in the study are included in the Appendix A, further inquiries can be directed to the corresponding author.

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
