# Peer review of "Infraspecific Variation in Silene Seed Tubercles"

_plants, 2024, doi:10.3390/plants13101416_

Round 1

Reviewer 1 Report

Comments and Suggestions for Authors

1. Please explain how the seed tubercles for measurement from each seed were selected. I did not find an explanation in the Materials and Methods, but this is an important element of the research and a detailed description of the procedure should be included there.

2. Text line 82 and 83: "2.1. General morphological measurements" and " 2.1.1. Differences between subgenera based on tubercle measurements", but in my opinion it should say "... on seed measurements", because this chapter contains measurement results for seeds, not tubercles.

3. In Tables 1 to 4, Appendix A, Tables A1 to A8 and A9 to A16 for consideration and if space permits, but I would prefer if the minimum and maximum values of the individual parameters were also given there.

4. Text line 111 (table 2 heading): the sentence: "General morphological measurements".  Maybe to make it easier to understand and distinguish, I suggest adding that it concerns the measurements of seeds.

5. Text lines 118 and in Table 2: the N parameter is explained twice: in the table header and below the table. Please simplify it.

6. Text lines 161-165 (Table 3 heading): The N parameter and why there are fewer seeds than in Table 1 are not explained.

7. Text lines 500 and 501 (Appendix A, Tables A1 to A8): Please correct the sentence, the word "population" used twice makes it unnecessarily difficult to understand.

Author Response

Dear Reviewer,

Thank you very much for your comments on our article. Please find below the detailed responses to the issues raised by your review. The text has been corrected accordingly. 

  1. Please explain how the seed tubercles for measurement from each seed were selected. I did not find an explanation in the Materials and Methods, but this is an important element of the research and a detailed description of the procedure should be included there.

This important question has been now explained in the Materials and Methods section. We have added the following paragraph to this section (lines 434-437):

“To obtain accurate and reproducible measurements, well-defined and regular shaped tubers were selected. Small, irregular or densely compacted tubercles can make measurements of curvature, height, and width at the base difficult or confuse. “

  1. Text line 82 and 83: "2.1. General morphological measurements" and " 2.1.1. Differences between subgenera based on tubercle measurements", but in my opinion it should say "... on seed measurements", because this chapter contains measurement results for seeds, not tubercles.

Thank you for the annotation. The heading of these two sections has been corrected according to your commentary (lines 83 and 84).

  1. In Tables 1 to 4, Appendix A, Tables A1 to A8 and A9 to A16 for consideration and if space permits, but I would prefer if the minimum and maximum values of the individual parameters were also given there.

Maximum and minimum values have been added to all the tables as indicated.

  1. Text line 111 (table 2 heading): the sentence: "General morphological measurements". Maybe to make it easier to understand and distinguish, I suggest adding that it concerns the measurements of seeds.

Heading of Table 2 has been corrected following your indication (line 115).

  1. Text lines 118 and in Table 2: the N parameter is explained twice: in the table header and below the table. Please simplify it.

The footer of Table 2 has been deleted. It is true, as you indicated, that the information was repeated in the legend and in the footer to Table 2.

  1. Text lines 161-165 (Table 3 heading): The N parameter and why there are fewer seeds than in Table 1 are not explained.

The N parameter is now explained in the legend to Table 3. The number of seeds in this table is slightly lower to table 1 because not all the seeds used for general measurements were used for tubercle measurements. This is now indicated in the text (footer to Table 3 lines 170-171).

  1. Text lines 500 and 501 (Appendix A, Tables A1 to A8): Please correct the sentence, the word "population" used twice makes it unnecessarily difficult to understand.

Legend to tables A1 to A8 has been corrected.

Thank you for your comments, on behalf of the authors,

Emilio Cervantes

Reviewer 2 Report

Comments and Suggestions for Authors

This is a manuscript that reveals morphological information of interest since there is considerable ignorance about the subject matter.

The text is easy to read and understand. Its methodology is well planned and complete. An analysis of the results is carried out with interesting contributions and relevant news. I consider it to be an ideal working method for working in other genres that are also little studied.

Aspects that can be improved:

The figures should be larger since they cannot be seen well and you have a lot of value.
Likewise, the size of the seeds would have to be increased. I would put them next to each other to compare them better.

In abstract, I would highlight the novel results that have been obtained compared to previous works.

Author Response

Dear Reviewer,

Thank you very much for your comments on our article. Please find below the detailed responses to the issues raised by your review. The text has been corrected accordingly.

  1. The figures should be larger since they cannot be seen well and you have a lot of value. Likewise, the size of the seeds would have to be increased. I would put them next to each other to compare them better.

The size of figures has been increased. Similarly the size of seed images has been increased in these figures.

  1. In abstract, I would highlight the novel results that have been obtained compared to previous works.

The results of this work have been highlighted in the abstract.

Thank you for your comments, on behalf of the authors,

Emilio Cervantes

Corresponding author.

Round 2

Reviewer 1 Report

Comments and Suggestions for Authors

Dear Authors,

I have no further comments, I am satisfied with the changes introduced. Thank you and good luck.